# GENERATIVE MODELS FOR ALIGNMENT AND DATA EFFICIENCY IN LANGUAGE

## ABSTRACT

We examine how learning from unaligned data can improve both the data efficiency of supervised tasks as well as enable alignments without any supervision. For example, consider unsupervised machine translation: the input is two corpora of English and French, and the task is to translate from one language to the other but without any pairs of English and French sentences. To address this, we develop feature matching auto-encoders (FMAEs). FMAEs ensure that the marginal distribution of feature layers are preserved across forward and inverse mappings between domains. We show that FMAEs achieve state of the art for data efficiency and alignment across three tasks: text decipherment, sentiment transfer, and neural machine translation for English-to-German and English-to-French. Most compellingly, FMAEs achieve state of the art for semi-supervised neural machine translation with significant BLEU score differences of up to 5.7 and 6.3 over traditional supervised models. Furthermore, on English-to-German, FMAEs outperform last year's best models such as ByteNet (Kalchbrenner et al., 2016) while using only half as many supervised examples.

## 1    INTRODUCTION

Massive collections of supervised data have been essential to deep learning advances such as image classification (Krizhevsky et al., 2012), neural machine translation (Sutskever et al., 2014), and more recently, cross-domain and intra-domain alignments such as text-to-image synthesis (Reed et al., 2016) and image-to-image translation (Isola et al., 2017). However, perception-based domains most often arise without explicitly aligned pairs. Supervised examples are human-labelled, which presents a fundamental bottleneck in learning from natural images or language.

In this paper, we examine how learning from unaligned data can improve both the data efficiency of supervised tasks as well as enable alignments without any supervision. For example, consider unsupervised machine translation: the input is simply two corpora of English and French, and the task is to translate from one language to the other but without any pairs of English and French sentences. More generally in text, tasks often involve taking a source sentence as input and returning a target sentence with a shared representation as the input but with target-specific properties; other examples include text decipherment and "style transfer" of sentiment, authors, and/or genres.

Data efficiency and alignment have seen most success for dense, continuous data such as images. Namely, generative adversarial networks (GANs) and deep latent variable models have led to promising progress in semi-supervised image classification (Kingma et al., 2014; Salimans et al., 2016) and unaligned image-to-image translation (Zhu et al., 2017). However, there has been limited success for sparse, discrete data such as text. We outline two key challenges.

One challenge is in designing and training generative models of text. With GANs, an immediate problem is that the generator returns discrete output which prevents backpropagation for training. While recent work has considered workarounds such as policy gradients (Yu et al., 2017) and continuous relaxations (Zhang et al., 2016), no approach has shown convincing empirical success over well-tuned LSTMs (Melis et al., 2017). On the other hand, deep latent variable models face challenges in utilizing the latent code with a flexible decoder such as an LSTM. While there have also been attempts at this problem (Bowman et al., 2016; Chen et al., 2017), autoregressive models without latent variables remain state of the art for language modeling (Shazeer et al., 2017).

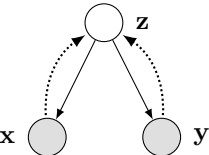

**Figure 1:** Latent variable model where $z$ represents invariant structure across domains. Dotted lines represent inference. With unaligned data, we only observe individual $x$'s and $y$'s and not pairs.

Another challenge arises in the task of alignment with few or even zero supervised examples: namely, it is open how to place inductive biases in our model and learning dynamics. For inductive biases in the learning dynamics, current approaches apply shared layers across forward and inverse mappings (Sutskever et al., 2015; Liu & Tuzel, 2016) and cycle consistency (Kim et al., 2017; Zhu et al., 2017; Yi et al., 2017). The latter is most successful but is not applicable to discrete data. For inductive biases in the model, convolutional filters prove crucial: unaligned image translation typically learns local morphologies such as the skin color of zebras and horses (Zhu et al., 2017). Unlike images, spatial invariance does not hold for language.

In this work, we develop FMAEs. FMAEs are motivated from a latent variable model which asserts that for each example, there exists a latent, structural code representing content that is shared across domains. Instead of a latent variable, FMAEs share the marginal distribution of feature layers across domains. This implicitly places an inductive bias based on constraints in the objective and network architecture, rather than explicitly with a prior. FMAEs involve training with an adversarial penalty which enables learning on a dense, lower-dimensional continuous feature space rather than directly on the sparse, discrete data.

We show empirically that FMAEs enable powerful data efficiency and alignment across three tasks: text decipherment, sentiment transfer, and neural machine translation. FMAEs are state of the art on all three. Most compellingly, FMAEs achieve state of the art for semi-supervised neural machine translation with significant BLEU score differences of up to 5.7 and 6.3 over traditional supervised models. Furthermore, on English-to-German, FMAEs outperform last year's best models such as ByteNet (Kalchbrenner et al., 2016) while using only half as many supervised examples.

## 2 LATENT VARIABLE MODEL FOR ALIGNMENT

We formally describe the problem and then derive an approach from probabilistic models. There are two data sets of i.i.d observations $\{x\} \sim p_{\text{data}}(x)$ and $\{y\} \sim p_{\text{data}}(y)$. Each data point is a variable-length sequence of discrete values, $x = (x_1, \ldots, x_T)$, and similarly for $y$.[1] The goal is to learn a mapping between the domains $G : X \to Y$, or conversely, $F : Y \to X$. Given a test input in one domain, this lets us predict the output in the other.

With probabilistic generative models, a natural approach is to posit a generative process according to the factorization $p(z)p(x \mid z)p(y \mid z)$ (Figure 1). The latent variable $z$ has a fixed prior, and each domain's observations are drawn conditionally independent given $z$ via a neural network. This model has been studied as a principle for one shot learning and domain adaptation (Sutskever et al., 2015), and has also been revisited for multimodal learning (Suzuki et al., 2016; Wang et al., 2016; Higgins et al., 2017; Vedantam et al., 2017).

Under the maximum likelihood principle, we maximize the objective

$$\mathbb{E}_{p_{\text{data}}(x)}[\log p(x)] + \mathbb{E}_{p_{\text{data}}(y)}[\log p(y)]$$

with respect to model parameters. Maximizing is equivalent to minimizing the negative marginal density. Standard variational methods posit an upper bound on the loss (Jordan et al., 1999),

$$\mathbb{E}_{p_{\text{data}}(x)q(z \mid x)}[-\log p(x \mid z)] + \mathbb{E}_{p_{\text{data}}(y)q(z \mid y)}[-\log p(y \mid z)] +$$
$$\mathbb{E}_{p_{\text{data}}(x)}[\text{KL}(q(z \mid x) \,\|\, p(z))] + \mathbb{E}_{p_{\text{data}}(y)}[\text{KL}(q(z \mid y) \,\|\, p(z))]. \tag{1}$$

The first two terms in the objective are reconstruction errors which determine the average number of bits to capture a data point $x$ ($y$) under noisy encodings. The last two terms are divergences which

---

[1]For now, assume there are no supervised examples, which are paired observations $\{(x, y)\} \sim p_{\text{data}}(x, y)$.

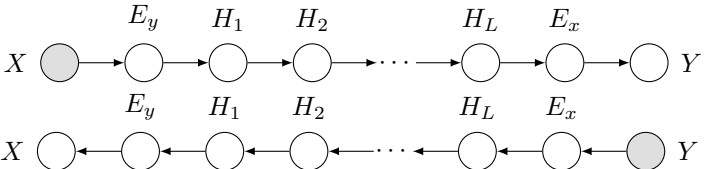

**Figure 2:** Feature matching auto-encoders (FMAEs). Each alignment mapping (top and bottom) is written as a composition of feature maps with embedding layers $E_x$ and $E_y$ and a sequence of hidden layers $H_\ell$. FMAEs minimize reconstruction error alongside an adversarial penalty which matches the marginal distributions over feature layers; in experiments, we only matched the embedding layers.

regularize the individual encoders; it shares information across domains via shrinkage toward the prior. After training, the model performs alignments by composing encoders with decoders: given an input $x$, map $x \to z$ via $q(z \mid x)$ and $z \to y$ via $p(y \mid z)$; the converse holds for an input $y$.

Empirically (§ 4), we found that a latent variable model trained with Equation 1 fails to learn alignments. From a statistical point of view, selecting the right prior is the key success for sharing across domains, and it is difficult to specify our assumptions about alignment this way. From a computational point of view, difficulties exist in optimizing the variational objective while utilizing the latent code. We consider alternatives using this model as motivation.

## 3  FEATURE MATCHING AUTO-ENCODER

The latent variable model in § 2 defines alignment mappings as a composition of encoder-to-decoders $X \to Z \to Y$ and $Y \to Z \to X$. Here, we consider alignment mappings under finer granularity as a sequence of feature maps. Figure 2 displays a composition of two embedding layers and $L$ hidden layers for each alignment mapping $G : X \to Y$ and $F : Y \to X$. For layer indices $\ell = 0, \ldots, L + 1$, denote individual feature maps as $g_\ell : H_{\ell-1} \to H_\ell$ for $G$ and $f_\ell : H_{\ell-1} \to H_\ell$ for $F$; $\ell = 0$ and $L + 1$ include the embeddings as domain and range respectively.

For a layer $\ell \in \{0, \ldots, L + 1\}$, consider the invariance property

$$p(g_\ell(x)) = p(f_{L+1-\ell}(y)). \tag{2}$$

Equation 2 says that the marginal distribution of a feature layer should be the same regardless of whether the layer is induced by the data distribution on $x$ (left hand side) or if the layer is induced by the data distribution on $y$ (right hand side).

If the number of hidden layers $L = 1$, this invariance reduces to matching the distribution of the middle layer in the mappings $X \to Z \to Y$ and $Y \to Z \to X$. This mimics the distribution invariance in § 2 where two KL divergences penalize deviations from a fixed prior distribution; however, Equation 2 posits a single divergence which penalizes deviation from each other. This pulls information across domains with an implicit, learnable density on the features. It indirectly posits a prior over the shared space without the need to specify a fixed, tractable prior density.

Unlike the latent variable model, feature matching also matches across arbitrary layers in a neural network. This lets us perform matching on feature layers closest to data space (namely, the embedding layers) while still avoiding the difficulties of adversarial training directly on discrete sequences. Note this also avoids the issue of latent code utilization in the decoder: matching embedding layers forces the decoder to use the encoder output in order to marginally match the distribution of the outputted embedding layer.

To enforce Equation 2, we apply an adversarial penalty jointly over the desired feature layers. Namely, to match the two embedding layers, consider the penalty

$$\mathbb{E}_{p_{\text{data}}(\mathbf{x})p_{\text{data}}(\mathbf{y})}[f(\mathbf{e}_x, \mathbf{e}_y)] - \mathbb{E}_{p_{\text{data}}(\mathbf{x})p_{\text{data}}(\mathbf{y})}[f(\mathbf{e}_x, \mathbf{e}_y)], \tag{3}$$

where in the first term, $\mathbf{e}_x$ is set via the input embedding layer $X \to E_x$ and $\mathbf{e}_y$ is set to the output embedding via $X \to E_x \to \cdots \to E_y$; in the second term, the converse holds where $\mathbf{e}_x$ is set to the output embedding via $Y \to E_y \to \cdots \to E_x$ and $\mathbf{e}_y$ is set via the input embedding layer $X \to E_x$.

With an infinite capacity discriminator, the max over $f$ is equal to a Wasserstein distance between the marginal embedding distributions (Kantorovich & Rubinstein, 1958).

Using the penalty of Equation 3, FMAEs minimize an objective with reconstruction terms,

$$
\begin{aligned}
&\mathbb{E}_{p_{\text{data}}(\mathbf{x})}[-\log p(\mathbf{x} \,|\, \mathbf{e}_y)] + \mathbb{E}_{p_{\text{data}}(\mathbf{y})}[-\log p(\mathbf{y} \,|\, \mathbf{e}_x)] + \\
&\lambda \Big( \mathbb{E}_{p_{\text{data}}(\mathbf{x}) p_{\text{data}}(\mathbf{y})}[f(\mathbf{e}_x, \mathbf{e}_y)] - \mathbb{E}_{p_{\text{data}}(\mathbf{x}) p_{\text{data}}(\mathbf{y})}[f(\mathbf{e}_x, \mathbf{e}_y)] \Big).
\end{aligned}
\tag{4}
$$

We minimize Equation 4 with respect to encoder-decoder parameters and we maximize it with respect to the discriminator $f$. Both enable backpropagation.

Inuitively, while the penalty of Equation 3 encourages that the marginal distribution of embedding layers match across domains, it can be easy to satisfy this by trivially returning a fixed distribution. The first two reconstruction terms prevent this by forcing the dual mappings to reconstruct each other up to the last feature layer.

Equation 4 can also be interpreted as a CycleGAN for text (Zhu et al., 2017), which is a method for unaligned image translation. It defines mappings $G : X \to Y$ and $F : Y \to X$ enforcing four properties: "cycle consistency" of $F(G(x)) = x$ and $G(F(y)) = y$ per data point, and matching output distributions of $p(G(x)) = p_{\text{data}}(y)$ and $p(F(y)) = p_{\text{data}}(x)$. FMAEs relax cycle consistency to hold up to inverting the embedding output rather than the data output, and it matches the marginal distributions over features. This enables adversarial training over a dense, lower dimensional continous space rather than directly on the discrete sequences.

## 3.1 SEMI-SUPERVISED LEARNING

FMAEs extend to learn from supervised examples in addition to unaligned (unsupervised) ones. This has the advantage of improving the data effiency of supervised problems by incorporating the massive amount of unaligned data in the real world. Namely, to learn from paired examples $\{(x, y)\} \sim p_{\text{data}}(x, y)$, we include the typical likelihood terms in the objective to encourage $G(x) = y$ and $F(y) = x$ per data point,

$$
\lambda_{\text{sup}} \, \mathbb{E}_{p_{\text{data}}(\mathbf{x}, \mathbf{y})}[-\log p(\mathbf{x} \,|\, \mathbf{y}) - \log p(\mathbf{y} \,|\, \mathbf{x})],
\tag{5}
$$

where $\lambda_{\text{sup}} \in \mathbb{R}^+$ balances how much we weigh aligned examples over unaligned examples.

An appealing property of FMAEs is that they formally handle the bridge between unsupervised and supervised learning. Strictly unaligned data results in unsupervised learning; unaligned and aligned data results in semi-supervised learning; and strictly aligned data results in supervised learning with dual mappings (Xia et al., 2017). FMAEs also extend to alignment over multiple domains. The divergence measures encouraging Equation 2 may hold pairwise or may be defined over multiple distributions (Garcia-Garcia & Williamson, 2012).

## 3.2 ARCHITECTURE: ATTENTION

For the network architectures, we primarily follow the Transformer of Vaswani et al. (2017), which has seen strong success for sequence-to-sequence modeling. The Transformer uses only attention layers for both the encoder and decoder; see Figure 3. The encoder applies an embedding layer to inputs followed by $L$ layers of self-attention. The decoder applies an embedding layer followed by $L$ alternations of a self-attention layer and a layer which attends over the encoder hidden states.

We make three adjustments which we found improved our experiments. First, we add noise to attention layers in order to sparsify the locations to attend to; we provide detail in the next subsection (§ 3.3). Second, we apply layer norm to the input of each residual block instead of afterwards. Third, we use learnable positional encodings rather than fixed sinusoidal embeddings as a way to impose ordering in the sequences (Gehring et al., 2017). Namely, the embedding layers take input elements $x = (x_1, \ldots, x_T)$, perform a table look-up to obtain its word embedding free parameters $w = (w_1, \ldots, w_T)$, and sums it with positional free parameters $(p_1, \ldots, p_T)$ to return the embedding, $e_x = (w_1 + p_1, \ldots, w_T + p_T)$.

For the discriminator, we also apply $L$ self-attention layers and clip weights following Arjovsky et al. (2017). As the matching distributions assume independence among features, we parameterize

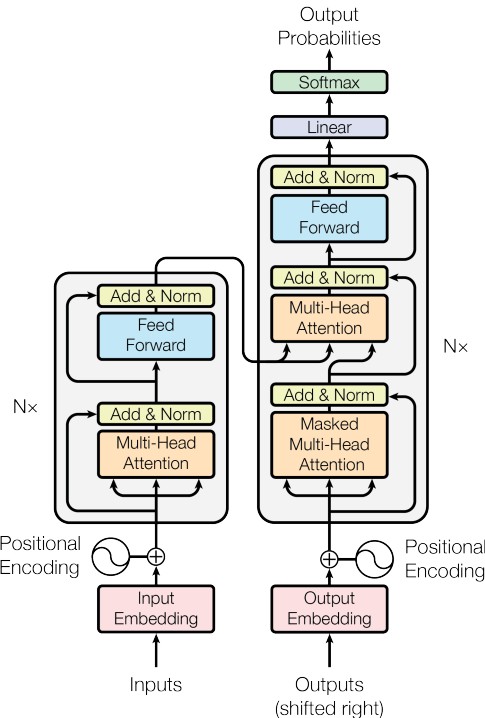

**Figure 3:** The Transformer network. We apply it twice with minor changes, once for each mapping in Figure 2. Figure from Vaswani et al. (2017).

the discriminator to not include interactions among its inputs. A problem when training Equation 4 is that we are matching the discriminator across two distributions with free parameters. This produces difficulties because the mappings can constantly scale the output of $f$ so long as the relative difference remains the same. To address this, we simply L2-normalize the inputs to $f$.

## 3.3 ADDING NOISE TO ATTENTION

An attention function can be described as taking a query and memory as input and returning a weighted sum over the memory states. The query and each memory state are vectors such as a decoder hidden state and the set of encoder hidden states respectively. This is also known as "soft attention," which is equivalent to taking an expectation over a categorical variable $z \in \{1, \ldots, T\}$ which attends to a specific memory state,

$$\mathbb{E}_{p(z \mid \mathbf{M}, q)}[\mathbf{M}_z] = \sum_{t=1}^{T} \pi_t \mathbf{M}_t,$$

where $z$'s distribution is a function of the matrix of memory states $\mathbf{M}$ and query vector $q$.

Soft attention produces dense weights where all states have a nonzero probability. Many tasks only require attending over few inputs such as machine translation, which often only requires finding the corresponding word to translate and its context. In order to sparsify the attended locations, we add noise to the softmax inputs: given a $T$-dimensional vector of logits inputs, return

$$\pi = \text{softmax}((\text{logits} + g)/\tau), \quad g = (g_1, \ldots, g_T), g_t \sim \text{Gumbel}(0, 1).$$

We use $\tau = 0.1$ in experiments. This forces the inputs to robustify against noise by taking on large positive or negative values (Frey, 1997). It is equivalent to a sample from the Gumbel-Softmax distribution and admits backpropagation (Jang et al., 2017; Maddison et al., 2017). It can be interpreted as a relaxation of "hard attention" (Xu et al., 2015), which requires score function gradients to handle the discrete variable. Adding noise augments each attention layer as a stochastic layer in a deep latent variable model; the temperature parameter bridges from hard to soft attention.[2]

---

[2] Empirically we find setting the temperature $\tau$ arbitrary close to 0 is undesirable; the model benefits from attending to few but multiple locations, whereas hard attention assumes one.

| Method | Substitution Cipher (BLEU) | | | | |
|---|---|---|---|---|---|
| | 20% | 40% | 60% | 80% | 100% |
| No transfer | 56.4 | 21.4 | 6.3 | 4.5 | 0 |
| Unigram matching | 74.3 | 48.1 | 17.8 | 10.7 | 1.2 |
| **Latent variable model (§ 2)** | 79.8 | 59.6 | 44.6 | 34.4 | 0.9 |
| **Latent variable matching** | 81.0 | 68.9 | 50.7 | 45.6 | 7.2 |
| Shen et al. (2017) | 83.8 | 79.1 | 74.7 | 66.1 | 57.4 |
| **Feature matching auto-encoder** | **90.3** | **84.2** | **78.7** | **65.4** | **60.5** |
| Supervision | 99.6 | 99.5 | 99.7 | 99.2 | 97.5 |

**Table 1:** BLEU scores for a word deciphering task. A varying percentage of words are substituted (ciphered) in each sentence. FMAEs outperform all methods across percentages.

## 4 EXPERIMENTS

We described the challenges of alignment for language and developed feature matching auto-encoders (FMAEs), an approach which ensures that the marginal distribution of feature layers are preserved across forward and inverse mappings. In experiments, we show that FMAEs enable powerful data efficiency and alignment across three tasks: text decipherment (§ 4.1), sentiment transfer (§ 4.2), and neural machine translation (§ 4.3). FMAEs are state of the art on all three. Most compellingly, FMAEs are not only state of the art for neural machine translation with limited supervision on EN-DE and EN-FR. They also outperform last year's best fully supervised models on EN-DE while using only half as many supervised examples.

In all experiments, we used the Adam optimizer with an initial step-size of one of $\{0.001, 0.0005, 0.0001\}$; we set $\beta_1 = 0.9$, $\beta_2 = 0.98$, and $\epsilon = 10^{-8}$. For machine translation, we followed the learning rate schedule of Vaswani et al. (2017) which increases the learning rate linearly for a fixed number of warm-up steps during training followed by a decrease proportional to the inverse square root of the step number. For text decipherment and sentiment transfer, we used a batch size of 256 unaligned input and output sequences. Sequences were batched together by approximate sequence length. Convolutional filters and weight matrices were initialized with Glorot uniform; embeddings initialized uniformly between $[-0.1, 0.1]$; biases initialized at 0.

### 4.1 TEXT DECIPHERMENT

For the first experiment, we provide intuitions behind alignment using a text cipher problem. We took Yelp restaurant reviews in 2017 and filtered out sentences exceeding 15 words. The training set consists of a corpus $X$ of 200,000 randomly selected sentences and a corpus $Y$ of a separate set of 200,000 sentences where we applied a word substitution cipher $f$. The cipher is a dictionary of key-value pairs which takes words in $X$ and returns a specific word in $Y$; the dictionary was generated at random and the two corpuses share the same vocabulary. We use test and validation sets of 100,000 supervised examples of paired sentences $\{(x, f(x)\}$.

Table 1 reports BLEU scores, a measurement of accuracy for sequence translations (Papineni et al., 2002), over levels of task difficulty. A fixed percentage of words in each sentence were replaced according to the cipher to generate the encrypted corpus $Y$. All models perform worse as the difficulty increases (higher percentage of substitution). The latent variable model of § 2 performs reasonably on low percentages of substitution but quickly worsens as the performance hinges on the latent structure. FMAEs outperform all methods, including the cross-aligned auto-encoder of Shen et al. (2017); this is the only recent work we're familiar with that has also studied this problem.

### 4.2 SENTIMENT TRANSFER

We also study sentiment transfer, a task which generally falls under the umbrella of "style transfer" in which the text's "content" is shared across domains and the "style" per domain is negative or positive sentiment. Unlike text decipherment which is a one-to-one mapping—each input has a

It is n't scary .
The script is smart and dark - hallelujah for small favors .

That 's its first sign of trouble .
' Blue Crush ' swims away with the Sleeper Movie of the Summer award .

Bring tissues .
This , sad nonsense .

It is depressing , ruthlessly pained and depraved , the movie equivalent of
 staring into an open wound .
Her performance moves between heartbreak and rebellion as she continually
 tries to accommodate to fit in and gain the unconditional love she seeks .

---

A solid , spooky entertainment worthy of the price of a ticket .
Of all the Halloween 's , this is the most visually unappealing .

Poetry in motion captured on film .
An imponderably stilted and self-consciously arty movie .

Steers refreshingly clear of the usual cliches .
A profoundly stupid affair , populating its hackneyed and meanspirited storyline
 with cardboard characters and performers who value cash above credibility .

A visual spectacle full of stunning images and effects .
An annoying orgy of excess and exploitation that has no point and goes nowhere .

**Table 2:** Sentiment transfer on Stanford Sentiment Treebank. **(top)** Negative to positive sentiment. **(bot)** Positive to negative sent. The top in each pair is test input; bottom is a draw from the model.

true (de)ciphered output—sentiment transfer is a many-to-many mapping in that there exist multiple plausible outputs for any input. This makes learning from unaligned data more plausible.

We use the Stanford Sentiment Treebank (Socher et al., 2013), which consists of roughly 8,500 sentences, 2,200 test sentences, 1,100 validation sentences, and a total of 21,700 word tokens. Each sentence has a sentiment label of 0-4. We partitioned the data into a corpus of positive sentiment (labels 0-1) and a corpus of negative sentiment (labels 3-4); we dropped neutral sentences. We initialized the embedding layers with GloVe embeddings and randomly elsewhere.

Table 2 displays sentiment transfer on sentences. The top in each pair of sentences is a unseen test input; the bottom represents a draw from the model. The results show powerful sentiment transfer: instead of trivial matching such as adjectives with their antonyms, the model indeed captures the general meaning of the sentence while stylizing it under a new sentiment. For example, it often does not use the same word during transfer but recognizes similar words in feature space: "scary" and "dark", "tissues" and "sad", "poetry" and "arty".

The second pair of sentences in Table 2 (top) shows an occasional failure mode where the model ignores the input and simply draws a random sentence that matches the output distribution. This happens when reconstruction error isn't precisely balanced with the adversarial penalty in the objective (Equation 4). Note an interesting effect is that because the training data is fairly small, the model overfits in that generations from the model can reproduce sentences it was trained on. For example, the sentence, "Of all the Halloween 's , this is the most visually unappealing ." exists in the training data. This means the alignment model essentially performed a nearest neighbors in feature space to find the closest sentence in content to the input but with opposite sentiment. (For the purpose of this experiment, this isn't a problem because we're more interested in the alignment transfer.)

### 4.3 Neural Machine Translation

For machine translation, we used the standard data sets of WMT 2014 EN-DE and WMT 2014 EN-FR. The WMT 2014 EN-DE data set consists of roughly 4.5 million sentence pairs. Following

| Method on EN-DE | # of Supervised Examples (BLEU) | | | |
| --- | --- | --- | --- | --- |
| | 500K | 1M | 2M | 4.5M |
| Transformer ($\approx$65M params) | 12.3 | 14.6 | 20.3 | |
| Transformer ($\approx$130M params) | 10.3 | 18.5 | 21.5 | |
| **Feature matching auto-encoder** | **16.0** | **21.4** | **24.0** | |
| Transformer ("big") (Vaswani et al., 2017) | | | | **28.4** |
| Conv Seq2Seq (Gehring et al., 2017) | | | | 25.16 |
| Google NMT (Wu et al., 2016) | | | | 24.6 |
| ByteNet (Kalchbrenner et al., 2016) | | | | 23.75 |
| RNN Enc-Dec-Att (Luong et al., 2015) | | | | 20.9 |
| RNN Enc-Dec (Luong et al., 2015) | | | | 14.0 |

| Method on EN-FR | # of Supervised Examples (BLEU) | | | |
| --- | --- | --- | --- | --- |
| | 500K | 1M | 2M | 36M |
| Transformer ($\approx$65M params) | 12.1 | 15.1 | 18.9 | |
| Transformer ($\approx$130M params) | 10.8 | 14.6 | 21.3 | |
| **Feature matching auto-encoder** | **17.1** | **23.1** | **26.7** | |
| Transformer ("big") (Vaswani et al., 2017) | | | | **41.0** |
| Conv Seq2Seq(Gehring et al., 2017) | | | | 40.46 |
| Google NMT (Wu et al., 2016) | | | | 39.92 |

Table 3: **(top)** BLEU scores on EN-DE newstest2014 test set while trained over a fixed number of supervised examples. **(bottom)** BLEU scores on EN-FR newstest2014 test set while trained over a fixed number of supervised examples. FMAEs outperform existing methods for semi-supervised translation with significant BLEU score differences.

Vaswani et al. (2017), we encoded sentences using byte-pair encoding, producing a shared source-target vocabulary of about 37,000 tokens. The WMT 2014 EN-FR data set consists of a much larger corpus of 36 million sentence pairs. We split tokens into a 32,000 word-piece vocabulary (Wu et al., 2016). To simulate a semi-supervised task, we partitioned both data sets into a fixed number of supervised pairs and made the rest unaligned. In both tasks, we evaluate performance with newstest2014 as test set and newstest2013 as validation set.

Table 3 displays results using FMAEs as well as current state-of-the-art translation models. For comparison on limited supervision, we also trained the Transformer network (with same modified architecture as the FMAE's); it only uses the available supervised examples.[3] In one version we used the same architecture and hyperparameters as one mapping in the FMAE (thus it has half the total number of parameters); in another version, we doubled the attention layer sizes to have comparable size in its single mapping to FMAE's dual mappings. The FMAE significantly outperforms the Transformer network. The BLEU scores have a dramatic difference from 2.5 and 5.4 on 2M supervised examples to up to 5.7 and 6.3 on 500K supervised examples.

Given the equivalent amount of supervision, the unaligned data set size in EN-FR (roughly 34-35M sentences per corpus) enables the FMAE to have improved BLEU scores in EN-FR over EN-DE. Most compellingly, we also note that our results for EN-DE outperformed last year's results of ByteNet (Kalchbrenner et al., 2016) while using only half as many supervised examples.

## 5 DISCUSSION

We developed feature matching auto-encoders (FMAEs), a method for learning from unaligned data in order to improve both the data efficiency of supervised tasks as well as to enable alignments without supervision. FMAEs achieved state of the art on three language tasks; most compellingly, FMAEs outperformed last year's best fully supervised models while using only half as many supervised examples. We believe our work is a compelling application of probabilistic generative models, where

---

[3]To the best of our knowledge, there are no approaches beyond FMAEs for semi-supervised machine translation.

data efficiency is crucial in supervised problems with high-dimensional input and output spaces. While we focused on neural machine translation, this is an important real-world problem for additional tasks such as summarization and on multimodal domains such as image captioning.

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
