# OpenReview forum: "Generative Models for Alignment and Data Efficiency in Language"
_ICLR.cc/2018/Conference — Reject_

### Official Review · AnonReviewer2 · 2017-11-27
**A little bit unclear**

**Rating:** 5
**Confidence:** 3

**Review:**

This work propose a generative model for unsupervised learning of translation model using a variant of auto-encoder which reconstruct internal layer representation in two directions. Basic idea is to treat the intermediate layers as feature representation which is reconstructed from the other direction. Experiments on substitution cipher shows improvement over a state of the art results. For translation, the proposed method shows consistent gains over baselines, under a condition where supervised data is limited.

One of the problems of this paper is the clarity.
- It is not immediately clear how the feature mapping explained in section 2 is related to section 3. It would be helpful if the authors could provide what is reconstructed using the transformer model as an example.
- The improved noisy attention in section 3.3 sounds orthogonal to the proposed model. I'd recommend the authors to provide empirical results.
- MT experiments are unclear to me. When running experiments for 2M data, did you use the remaining 2.5M for unsupervised training in English-German task?
- It is not clear whether equation 3 is correct: The first term sounds g(e_x, e_y) instead of f(...)? Likewise, equation 4 needs to replace the first f(...) with g(...).

---

### Official Review · AnonReviewer3 · 2017-11-27
**not good enough**

**Rating:** 4
**Confidence:** 3

**Review:**

This paper proposes a generative model called matching auto-encoder to carry out the learning from unaligned data.
However, it is very disappointed to read the contents after the introduction, since most of the contributions are overclaimed.

Detailed comments:
- Figure 1 is incorrect because the pairs (x, z) and (y, z) should be put into two different plates if  x and y are unaligned.

- Lots of contents in Sec. 3 are confusing to me. What is the difference between g_l(x) and g_l(y) if g_l : H_{l−1} → H_l and f_l: H_{l−1} → H_l are the same? What are e_x and e_y? Why is there a λ if it is a generative model?

- If the title is called 'text decipherment', there should be no parallel data at all, otherwise it is a huge overclaim on the decipherment tasks. Please add citations of Kevin Knight's recent papers on deciperment.

- Reading the experiment results of 'Sentiment Transfer' is a disaster to me. I couldn't get much information on 'sentiment transfer' from a bunch of ungrammatical unnatural language sentences. I would prefer to see some results of baseline models for comparison instead of a pure qualitative analysis.

- The claim on "FMAEs are state of the art for neural machine translation with limited supervision on EN-DE and EN-FR" is not exciting to me. Semi-supervised learning is interesting, but in the scenario of MT we do have enough parallel data for many language pairs. Unless the model is able to exceed the 'real' state-of-the-art that uses the full set of parallel data, otherwise we couldn't identify whether the models are able to benefit NMT.  Interestingly, the authors didn't provide any of the results that are experimented with full parallel data set. Possibly it is because the introduction of stochastic variables that prevent the models from overfitting on small datasets.

---

### Official Review · AnonReviewer1 · 2017-11-30
**The current paper is too sloppy to appear in a good conference: the concept is not described well and the experiments are not well-motivated**

**Rating:** 2
**Confidence:** 3

**Review:**


The paper is sloppily written where math issues and undefined symbols make it hard to understand. The experiments seem to be poorly done and does not convey any clear points, and not directly comparable to previous results.

(3) evaluates to 0, and is not a penalty. Same issue in (4). Use different symbols. I also do not understand how this is adversarial, as these are just computed through forward propagation.

Also what is this two argument f in eq 3? It seems to be a different and unspecified function from the one introduced in 2)

4.1: a substitution cipher has an exact model, and there is no reason why a neural networks would do well here. I understand the extra-twist is that training set is unaligned, but there should be an actual baseline which correctly models the cipher process and decipher it. You should include that very natural baseline model.

4.2 does not give any clear conclusions. The bottom is a draw from the model conditioned on the top? What was the training data, what is draw supposed to be? Some express the same sentiment, others different, and I have no idea if they are supposed to express the same meaning or not.

4.3: why are all the results non-overlapping with previous results? You have to either reproduce some of the previous results, or run your own experiment in matching settings. The current result tables show your model is better than some version of the transformer, but not necessarily better than the "big" transformer. The setup and descriptions do not inspire confidence.

Minor issues

3.1: effiency => efficiency

Data efficiency is used as a task/technique, which I find hard to parse. "Data efficiency and alignment have seen most success for dense, continuous data such as images."
"powerful data efficiency and alignment"

---

### Decision · Program_Chairs · 2018-01-29
**ICLR 2018 Conference Acceptance Decision**

**Decision:**

Reject

**Comment:**

The pros and cons of this paper cited by the reviewers (with a small amount of my personal opinion) can be summarized below:

Pros:
* The method itself seems to be tackling an interesting problem, which is feature matching between encoders within a generative model

Cons:
* The paper is sloppily written and symbols are not defined clearly
* The paper overclaims its contributions in the introduction, which are not supported by experimental results
* It mis-represents the task of decipherment and fails to cite relevant work
* The experimental setting is not well thought out in many places (see Reviewer 1's comments in particular)

As a result, I do not think this is up to the standards of ICLR at this time, although it may have potential in the future.